



# Improved model for correcting the ionospheric impact on bending angle in radio occultation measurements

Matthew J. Angling[1], Sean Elvidge[1], Sean B. Healy[2]

[1]Space Environment and Radio Engineering Group, University of Birmingham, UK.
[2]European Centre for Medium-range Weather Forecasts (ECMWF), Reading, UK.

*Correspondence to*: Matthew J. Angling (m.angling@bham.ac.uk)

**Abstract.** The standard approach to remove the effects of the ionosphere from neutral atmosphere GPS radio occultation measurements is to estimate a corrected bending angle from a combination of the L1 and L2 bending angles. This approach is known to result in systematic errors and an extension has been proposed to the standard ionospheric correction that is dependent

on the squared L1/L2 bending angle difference and a scaling term ($\kappa$). The variation of $\kappa$ with height, time, season, location and solar activity (i.e. the f10.7 flux) has been investigated by applying a 1D bending angle operator to electron density profiles provided by a monthly median ionospheric climatology model. As expected, the residual bending angle is well correlated (negatively) with the vertical TEC. However, $\kappa$ is more strongly dependent on the solar zenith angle. Furthermore, over the height range of interest (40-80 km) $\kappa$ is approximately linear with height. A simple $\kappa$ model has also been developed. It is

independent of ionospheric measurements, but incorporates geophysical dependencies (i.e. solar zenith angle, solar flux, altitude). The global mean error (i.e. bias) and the standard deviation of the residual errors are reduced to $-2.2\times10^{-10}$ rad and $2.0\times10^{-9}$ rad respectively. Although a fixed scalar $\kappa$ also reduces bias for the global average the selected value of $\kappa$ (14 rad$^{-1}$) is only appropriate for a small band of locations around the solar terminator. In the day time, the scalar $\kappa$ is consistently too high and this results in an over correction of the bending angles and a positive bending angle bias. Similarly, in the night time,

the scalar $\kappa$ is too low. However, in this case, the bending angles are already small and the impact of the choice of $\kappa$ is less pronounced.

## 1. Introduction

It has been demonstrated that, by using variational data assimilation techniques, GPS radio occultation (GPS-RO)
measurements can be assimilated into operational numerical weather prediction (NWP) systems to improve the accuracy of temperatures in the upper troposphere and lower/middle stratosphere (Healy & Thépaut 2006; Poli et al. 2009; Rennie 2010). In particular, GPS-RO measurements reduce stratospheric temperature biases in NWP systems and this indicates that such measurements could have an increasingly important role in climate monitoring and climate reanalyses (Poli et al. 2010; Steiner





et al. 2013). Notwithstanding the benefits of GPS-RO for the neutral atmosphere, it remains necessary to consider the effect of the ionosphere on the measurements.

(Vorob'ev & Krasil'nikova 1994) (hereafter referred to as VK94) proposed a method of combining the GPS-RO bending angles measured at two frequencies (L1 and L2) to provide a first order correction for the ionosphere. VK94 also showed that the first order correction leaves a systematic bending angle bias that increases as a function of the electron density squared, integrated over the vertical profile. The relationship between the bias and electron density suggests that the bending angle biases should vary diurnally and as a function of the 11-year solar cycle. This has been demonstrated by various authors; e.g. (Kursinski et al. 1997; Mannucci et al. 2011; Danzer et al. 2013).

(Healy & Culverwell 2015) have proposed a modification to the standard bending-angle correction to reduce the residual systematic ionospheric errors. The modification introduces a new second-order term that is a function of the square of L1 and L2 bending angle difference and a weighting term ($\kappa$). The aim of this work is to investigate the variation of $\kappa$ with height, time, season, location and solar activity (i.e. the f10.7 flux). This has been done by applying a 1D bending angle operator to electron density profiles provided by the NeQuick monthly median ionospheric climatology model (Nava et al. 2008). As well as examining the variations in $\kappa$, a $\kappa$ model has been developed. It is independent of ionospheric measurements, and therefore simple of implement in an operational system, but does incorporate the relevant geophysical dependencies (i.e. solar zenith angle, solar flux). The expectation is that, since NeQuick is a reasonable median model of the ionosphere, the $\kappa$ model derived from it will also exhibit reasonable statistics, though this has not been proven.

Radio occultations, the VK94 ionospheric correction procedure, and the proposed modified correction are described in Section 2. Examples of now $\kappa$ varies with height, location and solar activity are presented in Section 3. Models for $\kappa$ are proposed and assessed in Section 4 and the discussion and conclusions are given in Section 5.

## 2. Radio occultation and ionospheric corrections

(Kursinski et al. 1997) and (Hajj et al. 2002) provide a comprehensive description of the GPS-RO technique. In summary, the GPS satellites transmit on two L-band channels (L1, L2) at $f_1 = 1575.42$ MHz and $f_2 = 1227.60$ MHz and the signals are received by a satellite in low earth orbit (LEO) (Figure 1).

Assuming spherical symmetry, the bending angle of the ray between the GPS satellite and a receiver in LEO is:

$$\alpha_{Li}(a) = -2a \int_{r_t}^{\infty} \frac{dn_i/dr}{n_i\sqrt{(n_ir)^2 - a^2}} dr \qquad \text{Equation 1}$$





where $i = 1,2$ depending on the frequency; $a$ is the impact parameter; $r_t$ is the tangent height of the ray path; and $n_i$ is the refractive index. The impact parameter is given by:

$$a = nrsin(\phi) = const \qquad \text{Equation 2}$$

5    To a first order approximation, the refractive index comprises terms dependent on the neutral atmosphere refractivity ($N_n$), the ionospheric electron density ($n_e$), and the frequency ($f$) squared:

$$n_i \cong 1 + 10^{-6}N_n(r) - k\frac{n_e(r)}{f_i^2} \qquad \text{Equation 3}$$

where $k = 40.3 \text{ m}^3\text{s}^{-2}$. Therefore, the measured L1 and L2 bending angles are different from each other, and contain both neutral and ionospheric components. The standard approach taken in operational RO processing centres is to estimate a corrected neutral atmosphere bending angle ($\alpha_c$) using the VK94 approach:

$$\alpha_c(a) = \alpha_{L1}(a) + \frac{f_2^2}{f_1^2 - f_2^2}[\alpha_{L1}(a) - \alpha_{L2}(a)] \qquad \text{Equation 4}$$

where the L1 and L2 bending angles ($\alpha_{L1}$ and $\alpha_{L2}$ respectively) are interpolated to a common impact parameter. One benefit of this approach is that it is based on the standard parameters estimated by the RO retrieval system and does not require *a priori* information about the ionosphere. One downside is that a systematic bending angle error remains which increases as a

function of the electron density squared, integrated over the vertical profile. These residual ionospheric errors vary with the solar cycle and have been recognised as a potential source of bias in climatology products (Danzer et al. 2013). (Healy & Culverwell 2015) have proposed a modification to the standard ionospheric correction of the form:

$$\alpha_c(a) = \alpha_{L1}(a) + \frac{f_2^2}{f_1^2 - f_2^2}[\alpha_{L1}(a) - \alpha_{L2}(a)] + \kappa(a)(\alpha_{L1}(a) - \alpha_{L2}(a))^2 \qquad \text{Equation 5}$$

where the $\kappa$ term compensates for the systematic residual error in the standard approach. An appropriate value for $\kappa$ has been

investigated using simple analytic functions for the ionosphere (Healy & Culverwell 2015) and using a raytracer through a 3D ionospheric model (Danzer et al. 2015), though it should be noted that this study was limited to a low latitude band because of noise in the simulation system. It has been shown that $\kappa$ generally falls in the range of 10 to 20 rad$^{-1}$ and a simple scalar model, $\kappa\sim14$, provides a good first approximation, improving the accuracy of the "dry" temperature retrievals (Danzer et al. 2015). Nevertheless, it is clear that $\kappa$ will vary as a function of height, time, season, location and solar activity and therefore it is

possible that existing ionospheric climatology models could be used to compute an improved correction term by modelling the monthly mean, temporal and spatial variations of $\kappa$ more realistically.





## 3. Examples of $\kappa$ dependencies

A month median 3D ionospheric model (in this case NeQuick) and a 1D bending angle operator (based on Equation 1) can be used to estimate the residual ionospheric error and thereby estimate values for $\kappa$.

### 3.1. NeQuick

NeQuick is an monthly median ionospheric electron density model developed at the Aeronomy and Radiopropagation Laboratory (now Telecommunications/ICT for Development Laboratory) of the Abdus Salam International Centre for Theoretical Physics (ICTP), Trieste, Italy, and at the Institute for Geophysics, Astrophysics and Meteorology (IGAM) of the University of Graz, Austria (Nava et al. 2008). The model is based on the Di Giovanni - Radicella (DGR) model (Di Giovanni & Radicella 1990) which was modified for the PRIME project in COST 238 to provide electron densities from ground to 1000 km. The model has been designed to have continuously integrable vertical profiles which allows for rapid calculation of the TEC for trans-ionospheric propagation applications. The current versions NeQuick can be run up to a height of 20000 km and is used in the Galileo GNSS system to calculate ionospheric corrections (Angrisano et al. 2013).

NeQuick is a "profiler" which makes use of three profile anchor points at the E layer peak, the F1 peak, and the F2 peak. To specify the anchor points it uses the layer critical frequencies (foE, foF1, foF2) and the F2 maximum usable frequency factor (M3000(F2)) (Davies 1965). foE is determined using a solar zenith angle model; foF1 is assumed to be proportional to foE during daytime and zero during nighttime; and foF2 and M3000(F2) are derived from the ITU-R (CCIR) coefficients in the same way as the International Reference Ionosphere (IRI) (Bilitza & Reinisch 2008).

Between 100 km and the peak of the F2 layer, NeQuick uses an electron density profile based on the superposition of five semi-Epstein layers (Epstein 1930; Rawer 1983); i.e. the Epstein layers have different thickness parameters for their top and bottom sides. The topside of NeQuick is a simplified approximation to a diffusive equilibrium. A semi-Epstein layer represents the model topside with a height-dependent thickness parameter that has been empirically determined.

The model used in this work is the University of Birmingham's translation of the NeQuick v2.0.2 from FORTRAN into Python. Very minor (negligible) differences in results are observed due to the use of different interpolation routines. The Python code has been largely vectorised to increase the speed of operation. Some additional modifications have been made and are described in Table 1.

### 3.2. $\kappa$ estimation

In each of the examples shown in the following sections the same basic procedure has been followed to estimate the value of $\kappa$:

1. Use NeQuick to estimate a vertical profile of electron density



2.  Convert the electron density ($n_e$), to the refractive index ($n_i$) using the 1$^{st}$ order approximation ($n_i = 1 - 40.3 n_e / f_i^2$) for each frequency (L1 and L2)

3.  Estimate bending angle using the 1D observation operator for L1 and L2

4.  Form the VK94 corrected bending angle ($\alpha_c$).

Since no neutral atmosphere is included in the estimate of the refractive index, $\alpha_c$ should be zero if VK94 provides a perfect correction. Any non-zero values are representative of the residual ionospheric error ($\Delta\alpha$) which, from Equation 5, is modelled as:

$$\Delta\alpha = \kappa(a)(\alpha_{L1}(a) - \alpha_{L2}(a))^2 \qquad\qquad \text{Equation 6}$$

Since the bending angles are known, this can be rearranged to provide an estimate of $\kappa$ as a function of the impact parameter:

$$\kappa(a) = \frac{\Delta\alpha}{(\alpha_{L1}(a) - \alpha_{L2}(a))^2} \qquad\qquad \text{Equation 7}$$

The main area of interest for $\kappa$ estimation is between 40 and 80 km. It is in this region where the residual error from the ionospheric correction is likely to be a major contributor to the overall error budget of neutral atmosphere retrievals.

### 3.3.    Example height dependence

The Figure 2 to Figure 5 show two examples of the vertical electron density profile, the L1/L2 bending angles, the residual error and $\kappa$. The test parameters are given in Table 2. Over the height range of interest (40-80 km), Figure 5 shows that $\kappa$ is approximately linear, but its gradient is dependent on the local time.

### 3.4.    Geographic dependence

The geographic dependence of bending angle correction can be demonstrated by plotting maps of the TEC (Figure 6), residual bending angle (Figure 7) and $\kappa$ (Figure 8). In this case, the test parameters are given in Table 3. As expected, the residual bending angle is well correlated (negatively) with the vertical TEC. However, $\kappa$ appears to be more strongly dependent on the solar zenith angle.

### 3.5.    Solar cycle dependence

The solar cycle dependence of $\kappa$ has been investigated by estimating $\kappa$ at a tangent height of 60 km above London for each day over the last 60 years (Table 4). The results (Figure 9) show that $\kappa$ is negatively correlated with f10.7; i.e. $\kappa$ is low when the vertical TEC is large which occurs when f10.7 is high. Furthermore, the dynamic range of $\kappa$ is considerably smaller than that of the f10.7 (and hence TEC and bending angle), varying by a factor of approximately 50% compared to approximately 300% for f10.7



## 4. Models of $\kappa$

### 4.1. Introduction

Section 3 has presented examples of how $\kappa$ can vary spatially and with solar cycle. In this section, simple models of $\kappa$ will be

5   assessed in order to evaluate their potential to reduce the residual bending angle errors in the VK94 correction. Three models

will be considered:

- $\kappa$ equals zero (zero-$\kappa$); this represents the current situation with the unmodified VK94 correction
- $\kappa$ is a scalar (scal-$\kappa$); this is the approach proposed by (Healy & Culverwell 2015)
- $\kappa$ is a function of latitude, longitude, solar zenith angle and solar flux (func-$\kappa$).

In order the build the models a set of 25000 $\kappa$ estimates were generated from NeQuick using random drivers (uniformly

distributed over the ranges in Table 5). The true solar flux is used for each randomly selected day/year. A further independent

set of 25000 $\kappa$ estimates were also generated using the same random parameter ranges to act as a test data set.

### 4.2. Scalar $\kappa$

The random $\kappa$ values are shown in Figure 10. The median value is marked by the horizontal line and has value of 14 rad$^{-1}$.

This value is used as the scalar model.

### 1. Functional form $\kappa$

The aim of this model is to produce a very simple polynomial function that mimics some of the form of $\kappa$ that is not accounted

for by the scalar model. Figure 8 is suggestive that $\kappa$ is a function of solar zenith angle – this is a convenient parameter to use

since it embodies the position, local time and season. Figure 11, Figure 12 and Figure 13 show $\kappa$ as a function of solar zenith

angle, f10.7 and altitude respectively. Note that the solar zenith angle has been extended to $\pi$ radians to account for when the

sun is below the horizon. The figures indicate broadly linear dependencies in all cases; therefore the following model is

proposed:

$$\kappa = a + bf_{10.7} + c\chi + dh \qquad \qquad \text{Equation 8}$$

Where $f_{10.7}$ is the f10.7 flux (sfu), $\chi$ is the solar zenith angle (rad) and $h$ is the height above the ground (km); $a, b, c, d$ are

scalars to be found by fitting the model to the data.





The Python code curve_fit from the scipy.optimize package has been used to fit the model. The parameter results and the associated variances are shown in Table 6. A plot of the NeQuick estimated $\kappa$ compared to the func-$\kappa$ is shown in Figure 14. Figure 15 shows the geographic distribution of func-$\kappa$ at 12 UT in June and December at 60 km altitude and with an f10.7 of 150. These maps can be directly compared with those in Figure 8.

### 4.3. Bending angle error reduction
The second set of 25000 randomly distributed points has been used to assess the reduction in residual bending angle for each of the $\kappa$ models (zero-$\kappa$, scal-$\kappa$ and func-$\kappa$). Figure 16 shows a histogram of the residual bending angle errors for the full data set. The bending angle error statistics are in Table 7.

Both the scal-$\kappa$ and func-$\kappa$ results are an improvement over the zero-$\kappa$ results. In the case of the func-$\kappa$, both the standard deviation and the mean error (i.e. bias) of the residual errors is reduced by an order of magnitude. In the case of the func-$\kappa$, the mean error (i.e. bias) and the standard deviation of the residual errors are reduced to $-2.2\times10^{-10}$ rad and $2.0\times10^{-9}$ rad respectively. Although the scal-$\kappa$ also reduces bias for the global average, the geographic variation of $\kappa$ (shown in Figure 8

and Figure 15) makes it clear that the selected value of $\kappa$ (14 rad$^{-1}$) is, in fact, only appropriate for a small band of locations around the solar terminator. The effect of this is clear if the residual error statistics are considered for day time and night time separately.

Figure 17 and Figure 18 show histograms for residual bending angle for day and night respectively. In the day time, the scalar

$\kappa$ is consistently too high and this results in an over correction of the bending angles and a positive bending angle bias (Table 7). Similarly, in the night time, the scalar $\kappa$ is too low. However, in this case, the bending angles are already small and the impact of the choice of $\kappa$ is less pronounced (Table 7).

### 5. Conclusions

Using the random selection of vertical profiles from the NeQuick the median $\kappa$ has been shown to be 14 rad$^{-1}$ and this is therefore an appropriate value for $\kappa$ if it is to be represented by a single scalar. This value agrees well with the result from (Healy & Culverwell 2015) and is in the range suggested by (Danzer et al. 2015). Representing $\kappa$ as a scalar has the advantage of simplicity and is appropriate if re-processing centres are focused on ensuring that global average biases are removed. However, it has been demonstrated that such an approach can lead to significant differences in the residual bending angle bias

between day and night. In the day, the results indicate that the bending angle bias switches sign from $-3.3\times10^{-8}$ rad for no correction, to $+7.6\times10^{-9}$ rad for the scalar $\kappa$ correction.



This limitation can be overcome using the simple $\kappa$ function model. This approach does not require independent ionospheric measurements and so remains easy to implement. It should be noted that the $\kappa$ model is based on a monthly median ionospheric model. Whilst this is a starting point it will be necessary to work with climate re-processing centres to develop an effective validation strategies of the bending angle corrections. It would also be useful to assess the sensitivity of stratospheric climatologies to the bending angle bias and standard deviation bounds determined by this study. Furthermore, the magnitude of other error terms ( i.e. non-symmetry (Zeng et al. 2016)) should be assessed in light of these results.

## 6.    Acknowledgements

This work was undertaken as part of a visiting scientist study funded by the Radio Occultation Meteorology Satellite Application Facility (ROM SAF) which is a decentralised processing center under the European Organisation for the Exploitation of Meteorological Satellites (EUMETSAT). The original NeQuick Fortran code was provided by ITCP.

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





# Figures

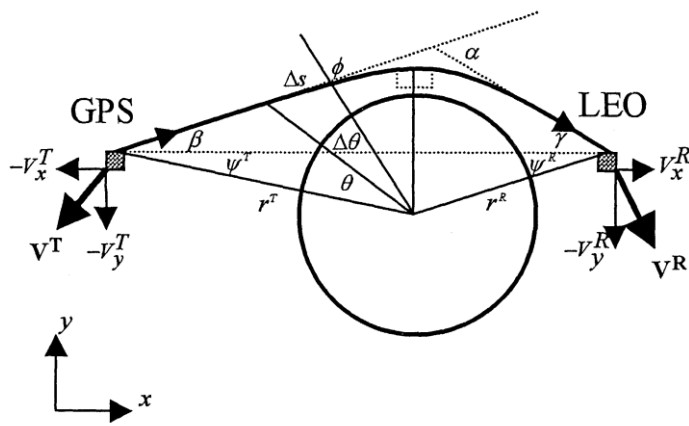

**Figure 1: Radio occultation geometry. Reproduced from** (Healy 2001)

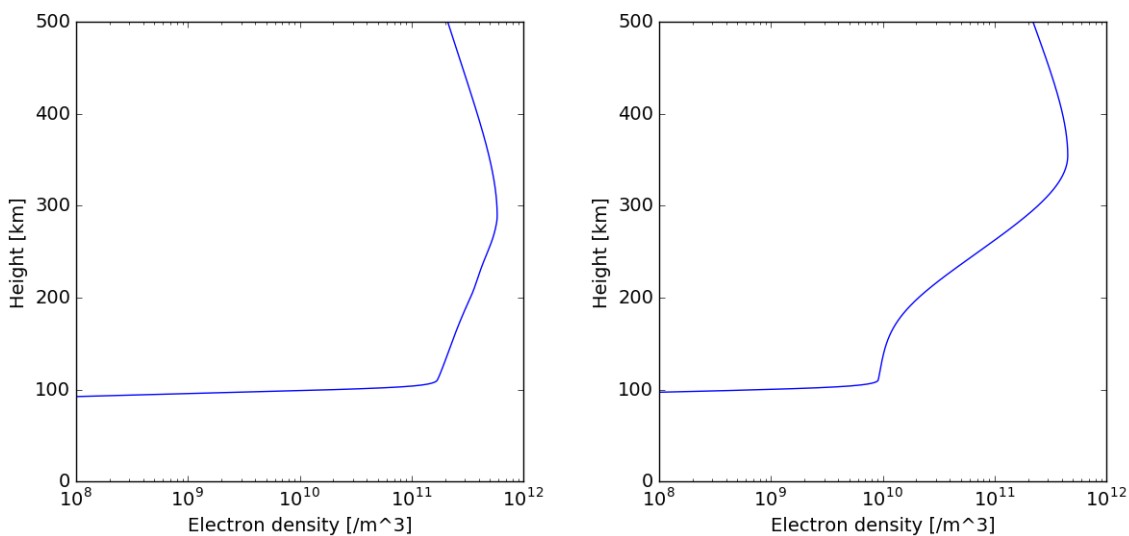

**Figure 2. Electron density profiles for test 1 (left, midday) and test 2 (right, midnight)**





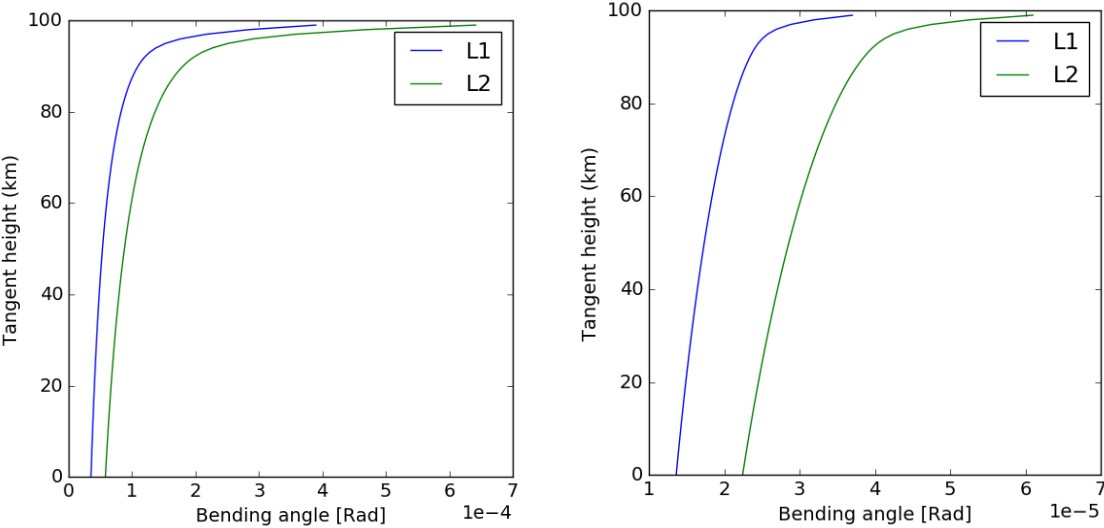

**Figure 3. L1 and L2 bending angles for test 1 (left, midday) and test 2 (right, midnight)**

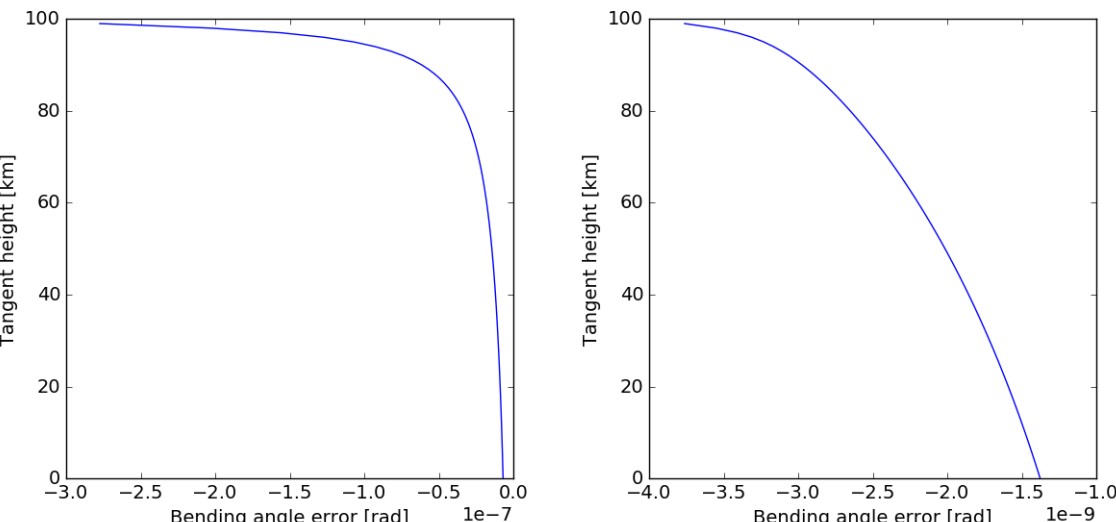

5    **Figure 4. Bending angle residual errors for test 1 (left, midday) and test 2 (right, midnight)**



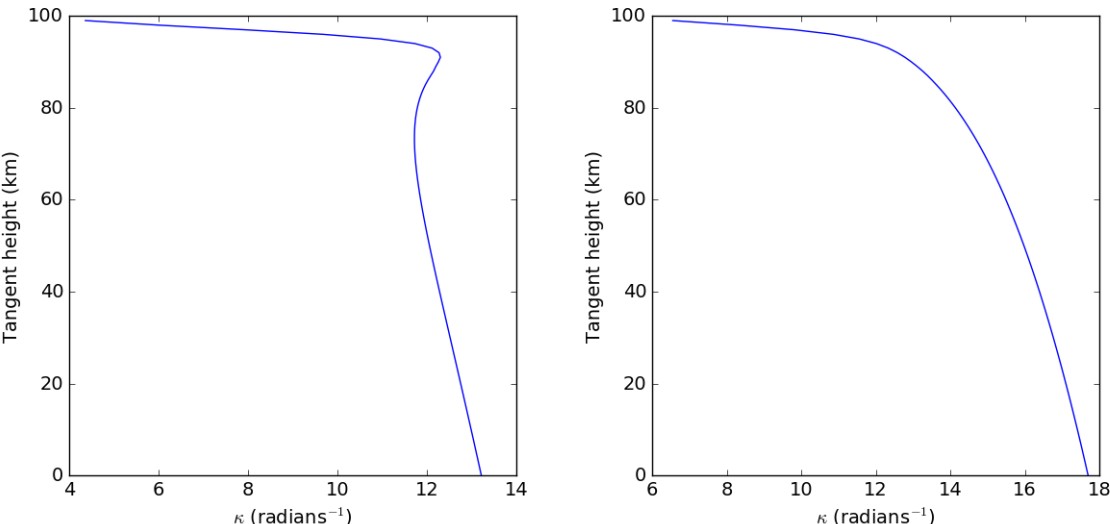

**Figure 5. Estimate of $\kappa$ for test 1 (left, midday) and test 2 (right, midnight)**

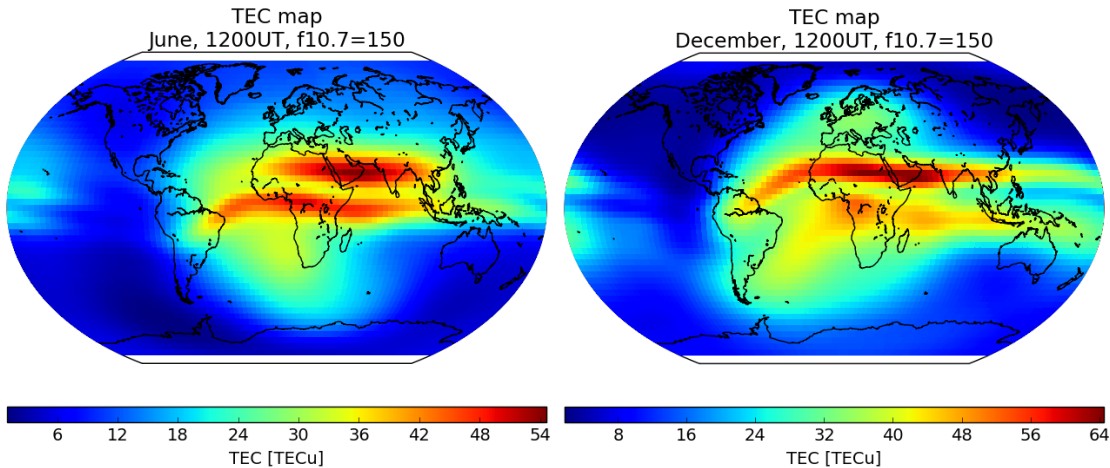

5    **Figure 6. Vertical TEC from NeQuick for 12 UT, f10.7=150, June (left) and December (right).**





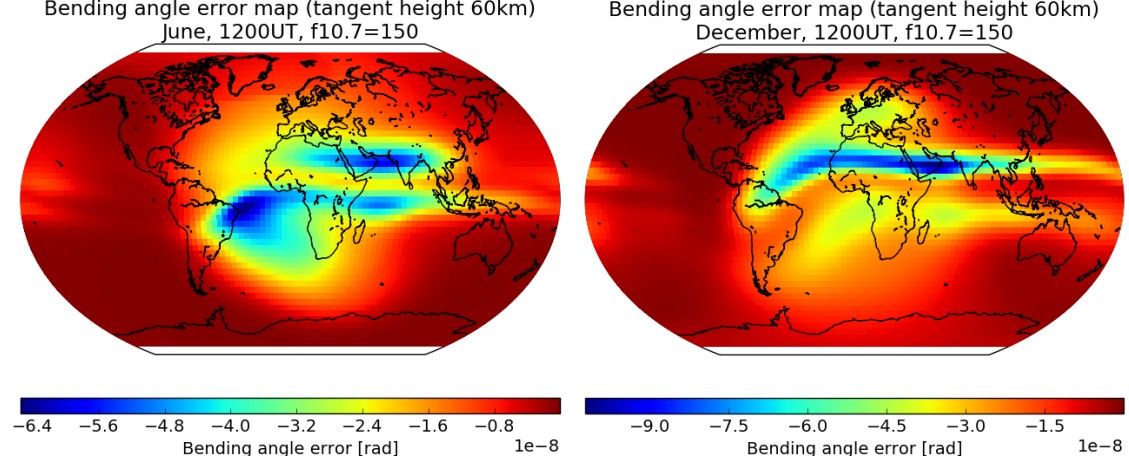

**Figure 7. Estimated residual bending angle error for 12 UT, f10.7=150, June (left) and December (right).**

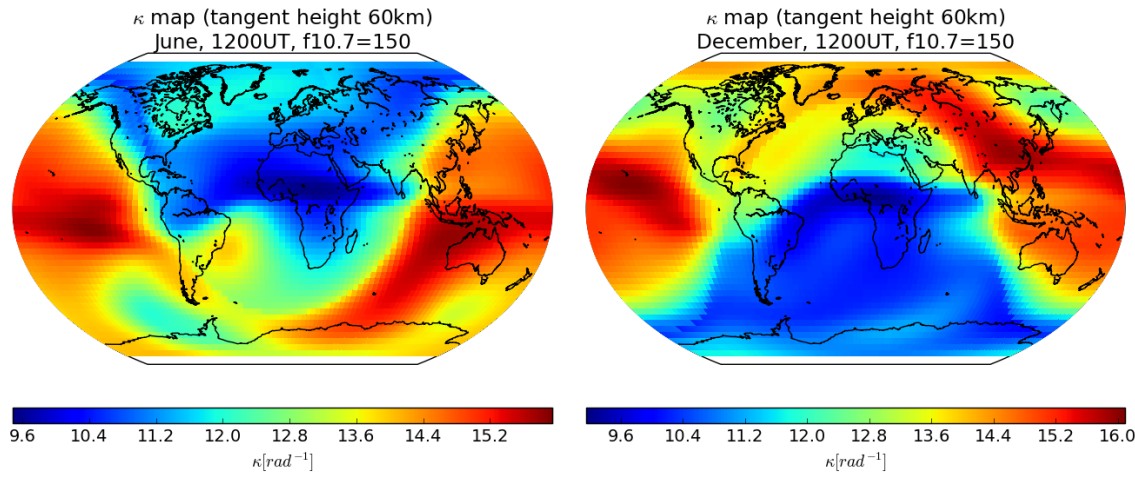

**Figure 8. Estimated $\kappa$ for 12 UT, f10.7=150, June (left) and December (right).**



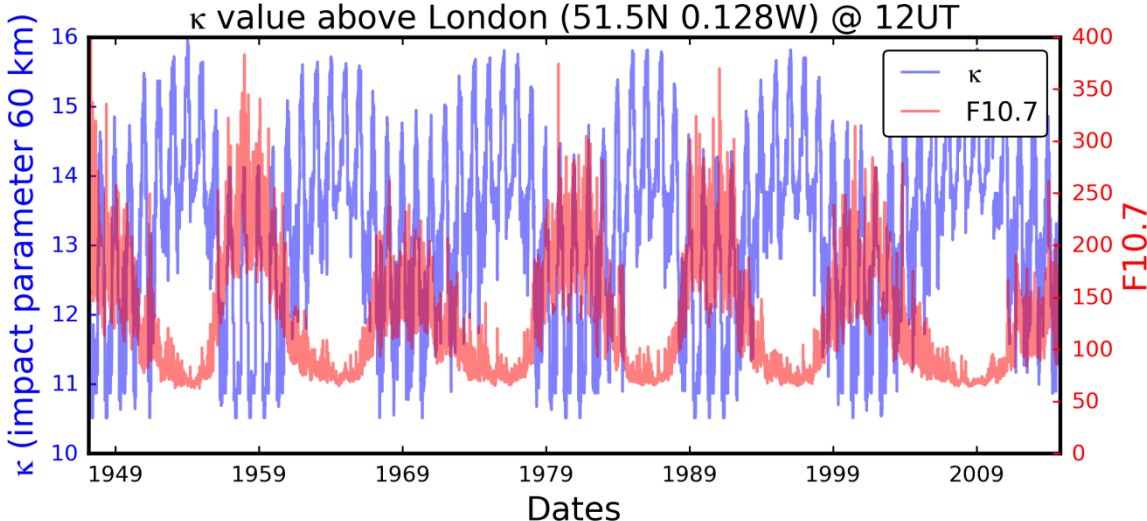

**Figure 9. Solar cycle dependence of $\kappa$ for a fixed location (London) and local time (12UT).**

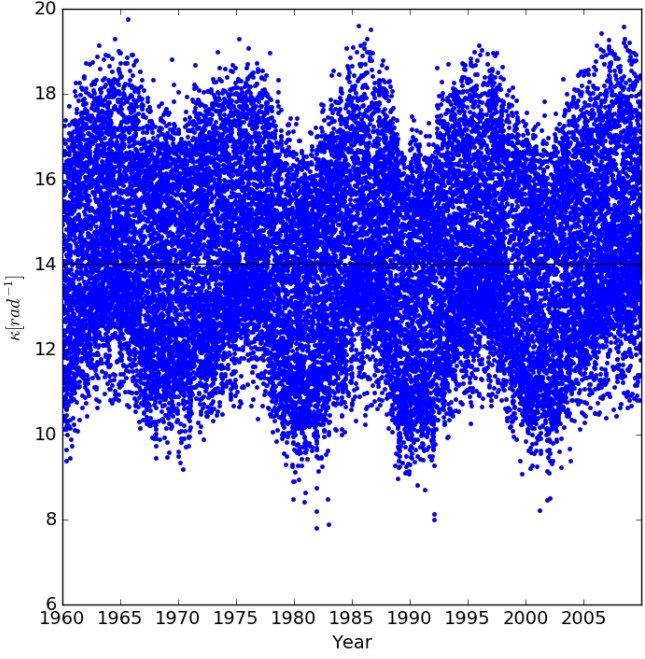

5    **Figure 10. $\kappa$ values for a random set of 25000 locations/times. The horizontal line marks the median (=14 rad$^{-1}$).**





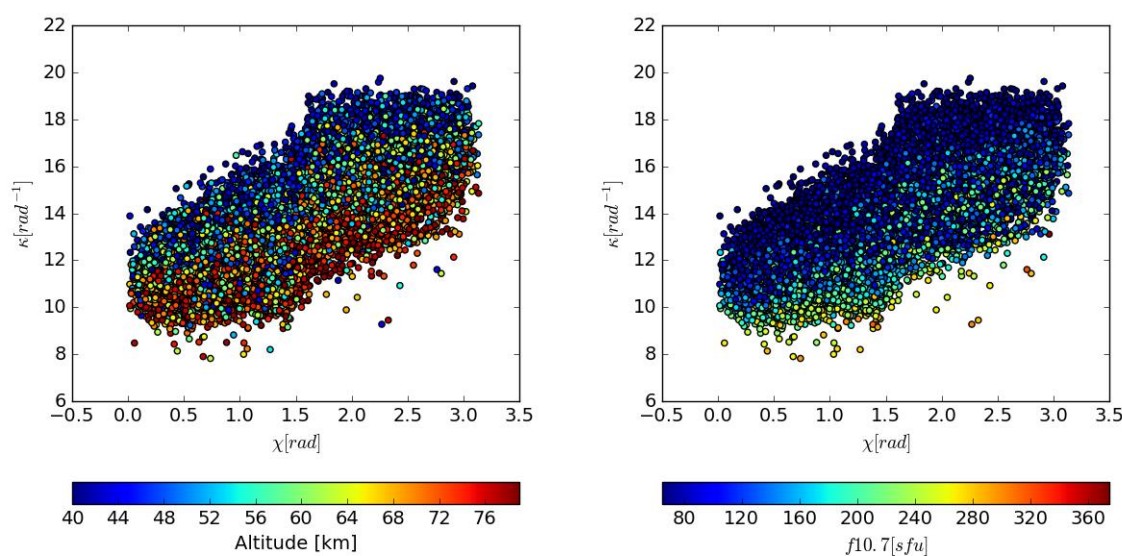

**Figure 11. $\kappa$ vs. solar zenith angle, colour coded by altitude (left) and f10.7 (right).**

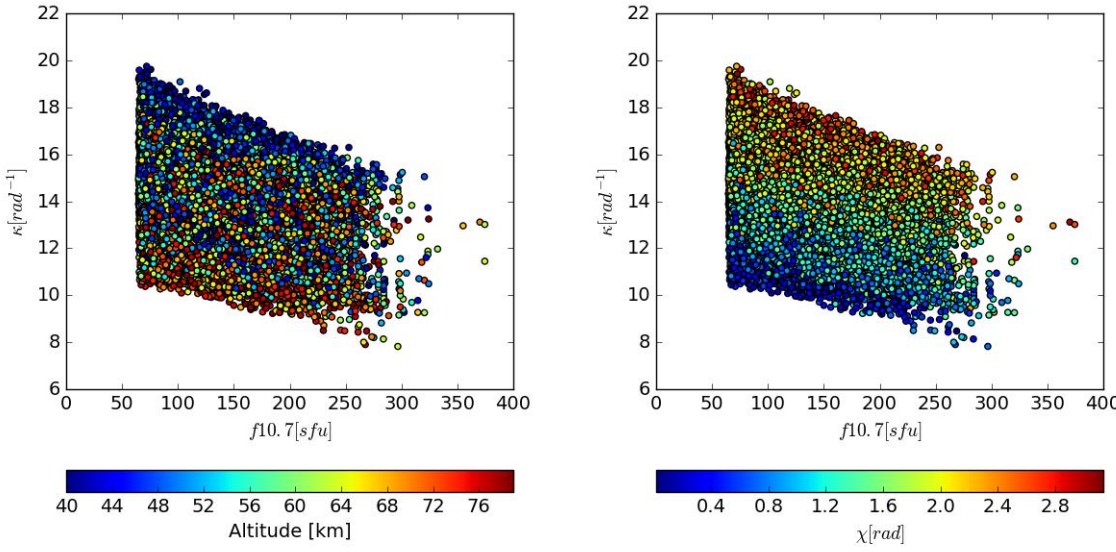

**Figure 12. $\kappa$ vs. f10.7, colour coded by altitude (left) and solar zenith angle (right)**



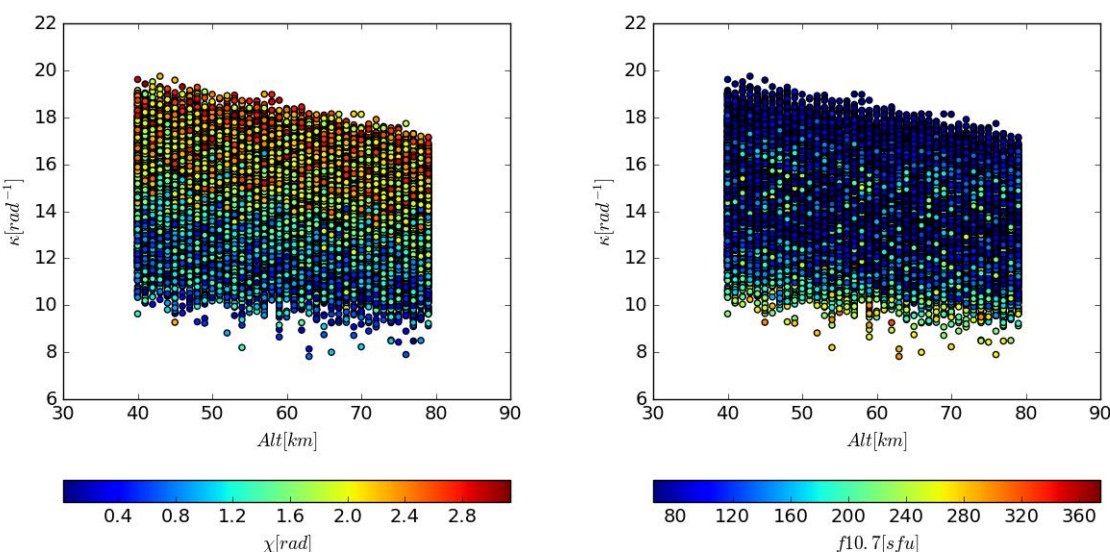

Figure 13. $\kappa$ vs. altitude, colour coded by solar zenith angle (left) and f10.7 (right)

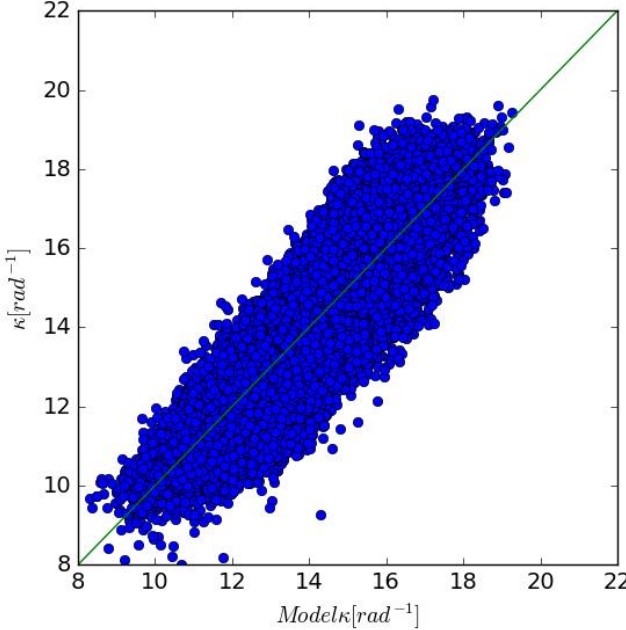

5    Figure 14. Scatter plot of $\kappa$ estimated from NeQuick compared to modelled $\kappa$.




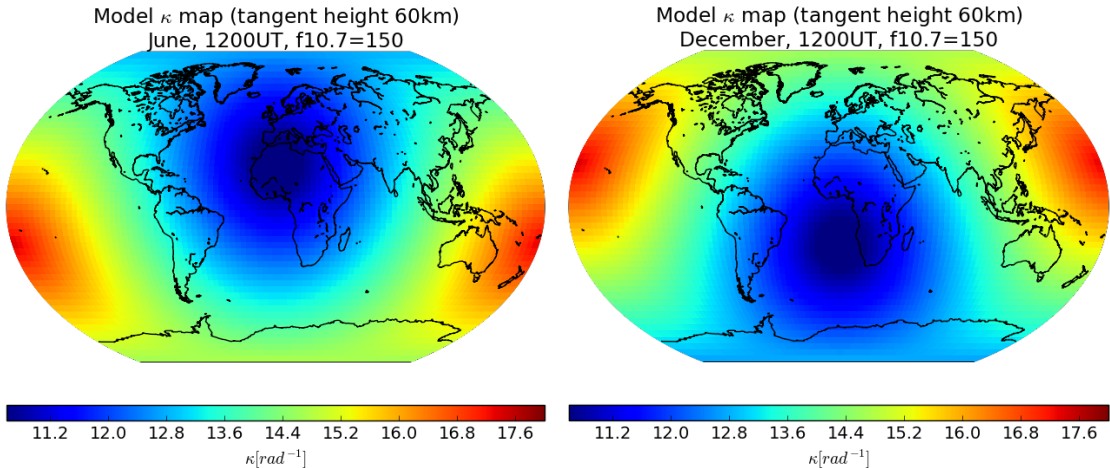

**Figure 15.** $\kappa$ model for 12 UT, f10.7=150, June (left) and December (right). c.f. Figure 8.

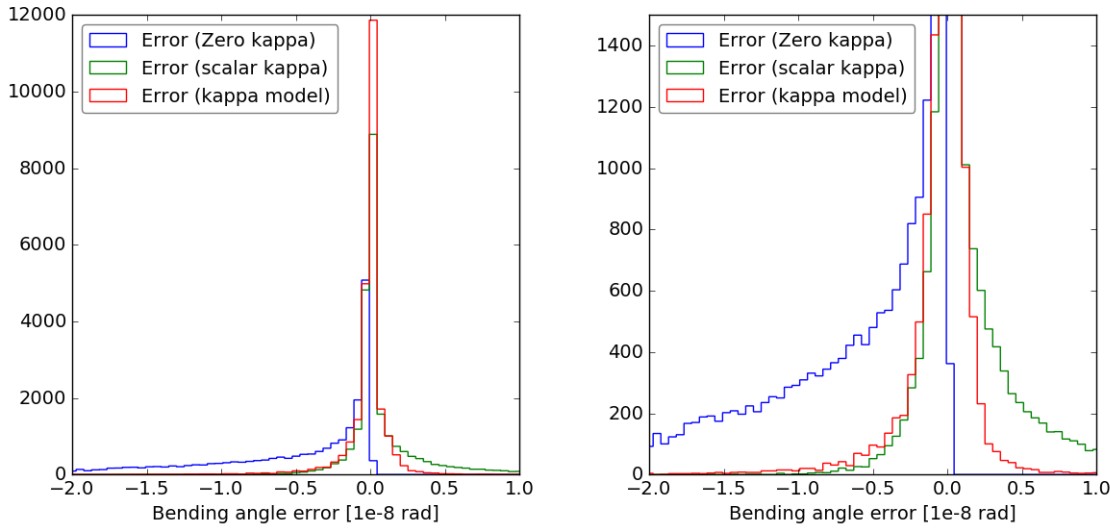

**Figure 16.** Histograms of globally distributed bending angle errors for zero $\kappa$, scalar $\kappa$, and modelled $\kappa$. Right: full histogram; left: zoomed to highlight tails.





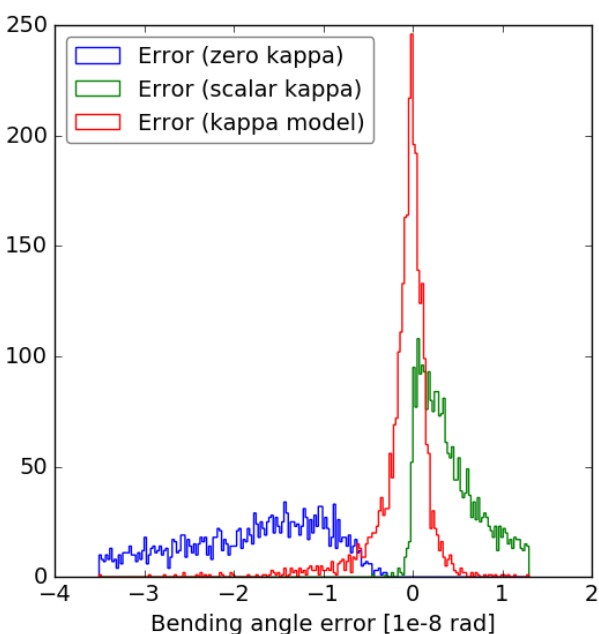

**Figure 17. Histograms of day time bending angle errors for zero $\kappa$, scalar $\kappa$, and modelled $\kappa$.**

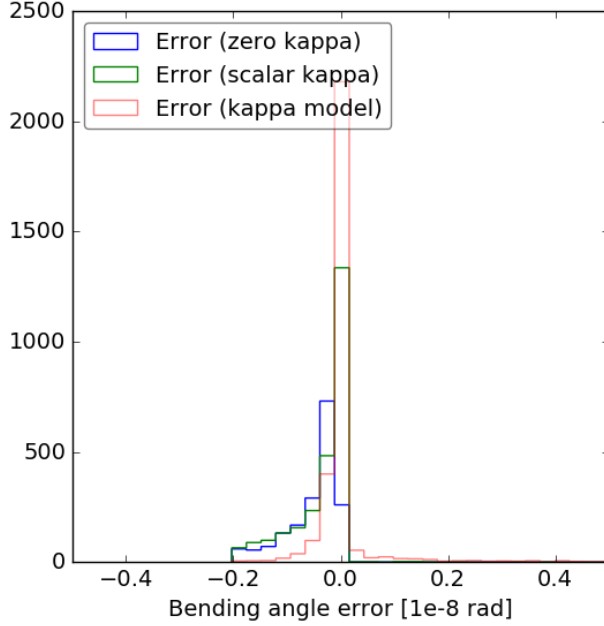

5    **Figure 18. Histograms of night time bending angle errors for zero $\kappa$, scalar $\kappa$, and modelled $\kappa$.**





**Tables**

| Feature | v2.0.2 | UoB variant |
|---|---|---|
| f10.7 | Clipped to:<br><br>63 < f10.7 < 193<br><br><br>This is the ITU recommendation for use with the ITU ionospheric coefficients | Clipped to:<br><br>63 < f10.7<br><br><br>Provides better TEC performance during high f10.7 solar cycle peaks. |
| Day of month | Not used | The day of month is used to linearly interpolate between two monthly coefficient files. This prevents step changes in electron density at month boundaries |
| hmE | Hard coded to 120 km | Hard coded to 110 km. This is a more reasonable value. However, a more sophisticated model will be implemented in future; i.e. (Chu et al. 2009). |
| Bottom side taper | Displays a discontinuity at 90 km that can produce artefacts in bending angle estimations | Bottomside taper added using a tanh function. |

**Table 1. Updates to produce the UoB variant of NeQuick.**

| Parameter | Test 1 | Test 2 |
|---|---|---|
| Latitude | 50° | 50° |
| Longitude | 0° | 0° |
| Time | 12 UT | 00 UT |
| Month | June | June |
| f10.7 | 150 | 150 |

**Table 2. Test parameters for height dependence examples**





| Parameter | Test 1 | Test 2 |
|---|---|---|
| Latitude | -85 to 85° | -85 to 85° |
| Longitude | -180 to 180° | -180 to 180° |
| Time | 12 UT | 12 UT |
| Month | June | December |
| f10.7 | 150 | 150 |
| Tangent height | 60 km | 60 km |

**Table 3. Geographic test parameters.**

| Parameter | Value |
|---|---|
| Latitude | 51.5° |
| Longitude | -0.128° |
| Time | 12 UT |
| Tangent height | 60 km |

**Table 4. Solar cycle test parameters.**

| Parameter | Range |
|---|---|
| Latitude | -80 to 80° |
| Longitude | -180 to 180° |
| Time | 0 to 23 UT |
| Day of year | 1 to 365 |
| Year | 1960 to 2010 |
| Tangent height | 40 to 80 km |

**Table 5. Parameter ranges for random $\kappa$ generation.**





| Parameter | Units | Estimated value | variance of the parameter estimate |
|-----------|-------|-----------------|-------------------------------------|
| a | rad$^{-1}$ | 15.05 | $1.764 \times 10^{-3}$ |
| b | rad$^{-1}$.sfu$^{-1}$ | $-1.243 \times 10^{-2}$ | $1.786 \times 10^{-8}$ |
| c | rad$^{-2}$ | 2.372 | $1.099 \times 10^{-4}$ |
| d | rad$^{-1}$.km$^{-1}$ | $-5.332 \times 10^{-2}$ | $3.351 \times 10^{-7}$ |

**Table 6. Estimated model parameters and associated variances**

| Region | Model | Mean (rad) | Median (rad) | Standard deviation (rad) |
|--------|-------|------------|--------------|--------------------------|
| Global | zero-$\kappa$ | $-1.3 \times 10^{-8}$ | $-4.5 \times 10^{-9}$ | $2.2 \times 10^{-8}$ |
|  | scal-$\kappa$ (14) | $1.5 \times 10^{-9}$ | $3.6 \times 10^{-13}$ | $5.4 \times 10^{-9}$ |
|  | func-$\kappa$ | $-2.2 \times 10^{-10}$ | $5.6 \times 10^{-13}$ | $2.0 \times 10^{-9}$ |
| Day-time | zero-$\kappa$ | $-3.3 \times 10^{-8}$ | $-2.3 \times 10^{-8}$ | $2.9 \times 10^{-8}$ |
|  | scal-$\kappa$ (14) | $7.6 \times 10^{-9}$ | $4.2 \times 10^{-9}$ | $9.9 \times 10^{-9}$ |
|  | func-$\kappa$ | $-9.8 \times 10^{-10}$ | $-3.0 \times 10^{-10}$ | $3.4 \times 10^{-9}$ |
| Night-time | zero-$\kappa$ | $-7.9 \times 10^{-9}$ | $-1.0 \times 10^{-9}$ | $2.3 \times 10^{-8}$ |
|  | scal-$\kappa$ (14) | $-7.0 \times 10^{-10}$ | $-1.5 \times 10^{-10}$ | $2.1 \times 10^{-9}$ |
|  | func-$\kappa$ | $1.7 \times 10^{-10}$ | $6.2 \times 10^{-12}$ | $1.9 \times 10^{-9}$ |

**Table 7. Global, day-time and night-time bending angle errors for three models**

