# Peer review of "Improved model for correcting the ionospheric impact on bending angle in radio occultation measurements"

_Atmospheric Measurement Techniques, 2017_

## Referee Comment (RC1) · N. Jakowski (Referee) · 7 Aug 2017

General comments: The manuscript deals with a very interesting approach to further reduce ionospheric correction errors in neutral atmosphere GPS radio occultation measurements. The manuscript should be of interest for readers of Atmos. Meas. Techniques. The manuscript is well structured shortly reviewing the classical method referred to as VK94 and the improved approach published by Healy and Culverwell (2015) before describing the new approach. The suggested approach is practically an extension of the formula suggested by Healy and Culverwell (2015) where the classical equation is extended by a quadratic term of the linear combination of the bending

angles at L1 and L2 frequencies. The contribution of this term is optimized by a fixed factor $\kappa$. To better mitigate the remaining biases the new approach considers this factor as being variable. The related model for $\kappa$ depends on the solar activity level (F10.7), the solar zenith angle and altitude. The presented results based on simulations using the NeQuik model as background indicate a clear improvement. To further confirm the results another model like IRI should be used. The publication of the manuscript is recommended.

Specific comments: Nevertheless there remains a significant difference between day-time and nighttime corrections which indicates still some bias. As the authors point out the remaining biases at nighttime can be ignored in practical applications due to their smallness. From a pragmatic point of view this is certainly true but from a scientific point of view this is unsatisfactorily. So I recommend discussing the observed day-night difference in more detail trying to explain the difference and to give an outlook how to improve the approach. As a starting point for the discussion here is my view: Former studies on ionospheric refraction of transionospheric radio waves have shown that not only the ionization level but also the profile shape plays a significant role on refraction as shown e.g. by Jakowski et al. (1994) and Hoque and Jakowski (2008, 2010). So I guess that the current $\kappa$ model considers the shape of daytime profiles very well because the daytime ionization is usually much higher than at nighttime. However it is well known that the vertical electron density profiles at night have a much broader shape as at daytime due to the relative higher contribution of the plasmasphere (e.g. Jakowski et al. 2017). So the resulting bending effects at night differ considerably from those at daytime due to the high non-linearity of ionospheric refraction. To overcome this problem I suggest adding a local time dependence of $\kappa$ in equation 8. The argumentation that the solar zenith angle embodies the position, local time and season is not sufficient because the slab thickness depends strongly on LT but much less from the solar irradiance (cf. Jakowski et al., 2017). So a separation of these dependencies by adding a LT term in equation 8 should improve the results.

Technical corrections: P6, l19: there is only one subsection, therefore should be cancelled P6 equation 8: the term dh looks like an increment of h, perhaps the coefficient symbol could be changed P9, l4: GPS instead of Gps Figure 9: unit of $\kappa$ missing Figures 8, 10-15: Axis font size too small

References Jakowski N., F. Porsch, and G. Mayer, Ionosphere - Induced -Ray-Path Bending Effects in Precision Satellite Positioning Systems, SPN 1/94, 6-13, 1994 Hoque M. M. and N. Jakowski (2008), Estimate of higher order ionospheric errors in GNSS positioning, Radio Sci., 43, RS5008, doi: 10.1029/2007RS003817. Hoque M. M., Jakowski N. (2010) Higher order ionospheric propagation effects on GPS radio occultation signals, Advances in Space Research, doi:10.1016/j.asr.2010.02.013 Jakowski, N. und Hoque, M. M. und Mielich, J. und Hall, C. (2017) Equivalent slab thickness of the ionosphere over Europe as an indicator of long-term temperature changes in the thermosphere. J. Atmos. Solar-Terr. Phys., DOI: 10.1016/j.jastp.2017.04.008 ISSN 1364-6826

---

## Author Comment (AC1) · 16 Oct 2017

The authors thank Dr Jakowski for his review and for this recommendation that our paper should be published.

We also note that Dr Jakowski refers to our statement that remaining kappa biases at nighttime can be ignored in practical applications due to the smallness of the consequent bending angle errors. He goes on with an interesting discussion on the possible reason for a residual nighttime bias and we agree that a likely cause is the difference in vertical profile shape between day and night. However, we would make three points:

[Figure]

1. The comment concerning the small bending angle errors at night refers to the scalar kappa value, not to the func-kappa. The functional kappa exhibits very little bias difference from day to night and the resultant bending angles errors can be seen to be reasonable symmetric about zero for day and night (fig17 and 18)

2. We have tried including an explicit dependency on local time in the model. It does not provide any additional model skill, suggesting that it does not do a better job of capturing the local time variability than the solar zenith angle.

3. It is important to remember that we are building our model by fitting kappa derived from NeQuick. NeQuick is based on the standard CCIR databases of foF2, foE and M3000F2, and therefore provides a reasonable median model of F and E regions' peaks. However, it is not certain that NeQuick is a good median representation of the layer shapes. Our approach, therefore, is to model kappa with minimal complexity to try to avoid a close fitting to NeQuick that may be inappropriate in reality. It would be fair to say that our results are indicative that a simple kappa model may be used, but that further testing with real data must be used to validate this. This will form the basis of a subsequent paper, but we have tried to be clearer in this point in the discussion section.

Technical corrections: P6, l19: there is only one subsection, therefore should be cancelled There was an error in the section numbering. This has been corrected

P6 equation 8: the term dh looks like an increment of h, perhaps the coefficient symbol could be changed Agreed. Changed to "e"

P9, l4: GPS instead of Gps Agreed. Now changed

Figure 9: unit of kappa missing Agreed. Now added

Figures 8, 10-15: Axis font size too small Agreed. Increased size

Please also note the supplement to this comment:
https://www.atmos-meas-tech-discuss.net/amt-2017-162/amt-2017-162-AC1-supplement.pdf

[Figure]

**Supplement:**

[revised manuscript text omitted]

---

## Referee Comment (RC2) · N. Jakowski (Referee) · 31 Oct 2017

I would like to thank the authors for the clarification I agree with.

There remains only a small correction to do in the supplement at page 2, line 25. Please modify "Kursinski et al. (1997) and Hajj et al. (2002) provide . . ."

---

## Referee Comment (RC3) · Anonymous Referee #2 · 6 Dec 2017

**Review of the paper "Improved model for correcting the ionospheric impact on bending angle in radio occultation measurements" by M. Angling, S. Elvidge and S. Healy.**

The authors consider residual error of the standard ionospheric correction, linear combination of L1 and L2 bending angles, related to ray separation at L1 and L2 GPS frequencies. In previous publications, this second order effect was approximated by squared L1-L2 bending angle with the coefficient. In the reviewed paper, the authors come up with the global model of the fitting coefficient and demonstrate that such model results in more effective reduction of the residual ionospheric error than constant coefficient. The results may be useful for climate applications of the GPS RO. I recommend publishing the paper after revision with account for comments below.

In this study, the authors: (i) assume local spherical symmetry of electron density; (ii) neglect higher order terms in the Appleton-Hartree equation. It may be useful to introduce these approximations at the beginning of the paper (currently (i) is mentioned in the last sentence of conclusions, while (ii) is not mentioned). Also, it may be useful to include reference to the paper by Hardy et al. (this paper may be available from different sources, see information at the end of the review). The paper by Hardy et al. also includes references to earlier publications on the second order ionospheric effects.

p.1, lines 13-14; p.5, lines 13-14: "*The main area of interest for k estimation is between 40 and 80 km. It is in this region where the residual error from the ionospheric correction is likely to be a major contributor to the overall error budget of neutral atmosphere retrievals.*"
First, why 40-80 km is the region of interest? I believe that for weather and climate applications, GPS RO may be somewhat useful at 40 km but it is totally useless at 80 km. An explanation or reference is needed. Second, "likely" means the authors are not sure that large-scale ionospheric residual is the major error contributor. An explanation or reference would be helpful.

p.1, line 13: "*As expected, the residual bending angle is well correlated (negatively) with the vertical TEC. However, k is more strongly dependent on the solar zenith angle.*"
In the context of the first sentence, the second sentence is not clear. In the approximation used by the authors, k depends only on electron density. The electron density, in turn, depends on the solar zenith angle. Thus k depends on the solar zenith angle through the electron density, and the expression "more strongly dependent" is not clear, unless it is explained "more strongly than what (?)". Also, see comment to p.5, line 24.

p.1, lines 16-17: "*The global mean error (i.e. bias) and the standard deviation of the residual errors are reduced to -2.2x$10^{-10}$ rad and 2.0x$10^{-9}$ rad respectively.*"
First, the number to which something is reduced requires the number from which that something is reduced. Second, it is not clear from the context, whether the reduction is relative to k=0 or k=const? The abstract should be self-explanatory. Also, see comment to p.7, line 12.

p.2, line 17: "... *simple of implement* ..."
It should be "simple to implement".

p.2, line 22: "*Examples of now k varies with height* ..."
It should be "how" instead of "now".

p.3, line 1: "$r_t$" is introduced but never used. Is it needed?

p.3, lines 14-15: "... *bending angle error* ... *which increases as a function of the electron density squared, integrated over the vertical profile*."
This sentence is not clear in several respects.
First, it is said "increases", but not said with what parameter? "Increases as a function" does not make sense (function may increase or decrease).
Second, "integration over" is commonly used with respect to domain (e.g., over height interval). Integration "over the profile" is not a common expression.
Third, if the authors mean equation (22) from VK94, it is more complicated than just integrated squared electron density; it includes derivative and kernel.
This sentence should be made clear and reference provided.

p.3, line 24: "... *as a function of ... time* ..."
Logically, it should be "local time".

p.4, line 2: "*A month median* ..."
It should be "A monthly median".

p.4, line 10: "PRIME" and "COST 238" should be explained.

p.4, line 12: "... *current version NeQuick* ..."
It should be "current version of NeQuick".

p.4, line 13: "... *Galileo GNSS system* ..."
In the "GNSS", the last "S" already stands for "system". The expression above should be corrected and "GNSS" explained.

p.5, line 16: "*Example height dependence*"
It should be "Example of height dependence"

p.5, line 19: "... *k is approximately linear* ..."
Linear with what parameter?

p.5, line 24: "... k *appears to be more strongly dependent* ..."
More strongly than what? Also, see comment to p.1, line 13.

p.5, lines 30-32: What is the physical sense of the statement that dynamic range of k is smaller than of F10.7? What are practical conclusions from this statement? This should be explained, otherwise I don't see why is this statement needed.

p.6, line 11: "*In order the build the models...*"
It should be "in order to build the models".

p.6, lines 20-25: It may be better to introduce "chi" here (rather than after equation 8) because "chi" is used in discussion of figures 11-13.

p.6, lines 11-13; p.7, line 7: What is the reason for using two different sets, generated with the same "random drivers", for building and testing the k-model? I assume that the sets are statistically representative and the results are statistically significant. Will the results be substantially different with the use of one set for building and testing?

p.7, line 12: "*... residual errors are reduced to 2.2x$10^{-10}$ rad and 2.0x$10^{-9}$ rad respectively*."
The number to which something is reduced requires the number from which that something is reduced. Also, see comment to p.1, lines 16-17.

Figure 9: Axes labels on all figures, except Figure 9, have units in parentheses. Thus "k (impact parameter 60 km)" is confusing. It should be "k (1/rad)". The title above the figure can be changed to "k value at 60 km above London".

Figure 16: I think, "left" and "right" are mixed up in the caption. Left is full histogram, while right is zoomed to highlight tails.

**Reference:**

K.R. Hardy, G.A Hajj, E.R. Kursinski: Accuracies of Atmospheric profiles Obtained from GPS Occultations.

(i) Proceedings of the  6th International Technical Meeting of the Satellite Division of ION (ION GPS 1993), Salt Lake City, UT, September 22-24, 1993, pp. 1545-1556.

(ii) International Journal of Satellite Communications and Networking, Vol. 12, No. 5, 1994, pp. 463-473.

---

## Author Comment (AC3) · 5 Jan 2018

**Response to Anonymous review of the paper "Improved model for correcting the ionospheric impact on bending angle in radio occultation measurements" by M. Angling, S. Elvidge and S. Healy.**

Responses are included in line below in red.

The authors consider residual error of the standard ionospheric correction, linear combination of L1 and L2 bending angles, related to ray separation at L1 and L2 GPS frequencies. In previous publications, this second order effect was approximated by squared L1-L2 bending angle with the coefficient. In the reviewed paper, the authors come up with the global model of the fitting coefficient and demonstrate that such model results in more effective reduction of the residual ionospheric error than constant coefficient. The results may be useful for climate applications of the GPS RO. I recommend publishing the paper after revision with account for comments below.

The authors thank the reviewer for her helpful comments and her recommendation to publish.

In this study, the authors: (i) assume local spherical symmetry of electron density; (ii) neglect higher order terms in the Appleton-Hartree equation. It may be useful to introduce these approximations at the beginning of the paper (currently (i) is mentioned in the last sentence of conclusions, while (ii) is not mentioned). Also, it may be useful to include reference to the paper by Hardy et al. (this paper may be available from different sources, see information at the end of the review). The paper by Hardy et al. also includes references to earlier publications on the second order ionospheric effects.

Agreed.
Reference to Hardy has been included. The following sentences have also been added to section 2:

"Horizontal gradients will result in residual errors in the inversion. However, it is expected that these errors are random; therefore, they should not affect monthly or seasonal climatologies."

"The first order approximation neglects terms involving higher powers of the frequency and the earth's magnetic field; however these have little effect on the residual bending angle errors (Syndergaard 2000)."

p.1, lines 13-14; p.5, lines 13-14: "*The main area of interest for k estimation is between 40 and 80 km. It is in this region where the residual error from the ionospheric correction is likely to be a major contributor to the overall error budget of neutral atmosphere retrievals.*"

First, why 40-80 km is the region of interest? I believe that for weather and climate applications, GPS RO may be somewhat useful at 40 km but it is totally useless at 80 km. An explanation or reference is needed. Second, "likely" means the authors are not sure that large-scale ionospheric residual is the major error contributor. An explanation or reference would be helpful.

A more detailed description of the useful vertical limits of k has been included in section 3.2

p.1, line 13: "*As expected, the residual bending angle is well correlated (negatively) with the vertical TEC. However, k is more strongly dependent on the solar zenith angle*." In the context of the first sentence, the second sentence is not clear. In the approximation used by the authors, k depends only on electron density. The electron density, in turn, depends on the solar zenith angle. Thus k depends on the solar zenith angle through the electron density, and the expression "more strongly dependent" is not clear, unless it is explained "more strongly than what (?)". Also, see comment to p.5, line 24.

We agree that this was unclear. The point is that whilst the residual bending angle error is strongly related to the TEC, the k is more strongly related to the solar zenith angle. This indicates that the TEC dependent component of the residual error is largely modelled effectively by the squared L1/L2 bending angle difference term in the correction. Thus, the k term is capturing other features such as the hmF2 variability. We have updated the text to make this clearer.

p.1, lines 16-17: "*The global mean error (i.e. bias) and the standard deviation of the residual errors are reduced to $-2.2 \times 10^{-10}$ rad and $2.0 \times 10^{-9}$ rad respectively*." First, the number to which something is reduced requires the number from which that something is reduced. Second, it is not clear from the context, whether the reduction is relative to k=0 or k=const? The abstract should be self-explanatory. Also, see comment to p.7, line 12.

Agreed. Text has been amended to compare results of the uncorrected case to the model k case

p.2, line 17: "*... simple of implement ...*" It should be "simple to implement".

Agreed. Text amended

p.2, line 22: "*Examples of now k varies with height ...*" It should be "how" instead of "now".

Agreed. Text amended

p.3, line 1: "$r_t$" is introduced but never used. Is it needed?

This is the lower limit in the equation 1 integral, so should be defined.

p.3, lines 14-15: "*... bending angle error ... which increases as a function of the electron density squared, integrated over the vertical profile*." This sentence is not clear in several respects.
First, it is said "increases", but not said with what parameter? "Increases as a function" does not make sense (function may increase or decrease).
Second, "integration over" is commonly used with respect to domain (e.g., over height interval). Integration "over the profile" is not a common expression.
Third, if the authors mean equation (22) from VK94, it is more complicated than just integrated squared electron density; it includes derivative and kernel. This sentence should be made clear and reference provided.

We agree that this was unclear. We have amended the text to read:
"One downside is that a systematic bending angle error remains (see equation 5 of Healy & Culverwell 2015). The bending angle error has a dependence on the electron density squared, which indicates that it will vary with the solar cycle. This has been recognised as a potential source of bias in climatology products (Danzer et al. 2013)."

p.3, line 24: "*... as a function of ... time ...*" Logically, it should be "local time".

Agreed. Text amended

p.4, line 2: "*A month median ...*" It should be "A monthly median".

Agreed. Text amended

p.4, line 10: "PRIME" and "COST 238" should be explained.

Agreed. Text amended.

p.4, line 12: "*... current version NeQuick ...*" It should be "current version of NeQuick".

Agreed. Text amended

p.4, line 13: "*... Galileo GNSS system ...*"
In the "GNSS", the last "S" already stands for "system". The expression above should be corrected and "GNSS" explained.

Agreed. Text amended

p.5, line 16: "*Example height dependence*" It
should be "Example of height dependence"

Modified to "Height dependence" for consistency with the following sections

p.5, line 19: "... *k is approximately linear* ..." Linear
with what parameter?

Tangent height. Text amended

p.5, line 24: "... k *appears to be more strongly dependent* ..."
More strongly than what? Also, see comment to p.1, line 13.

We agree that this was unclear. The point is that whilst the residual bending angle error is strongly related to the TEC, the k is more strongly related to the solar zenith angle. This indicates that the TEC dependent component of the residual error is largely modelled effectively by the squared L1/L2 bending angle difference term in the correction. Thus, the k term is capturing other features such as the hmF2 variability. We have updated the text to make this clearer:
"However, $\kappa$ is more strongly dependent on the solar zenith angle, indicating that the TEC dependent component of the residual error is largely modelled by the squared L1/L2 bending angle difference term in the correction, and that $\kappa$ is modelling other features such as changes in hmF2."

p.5, lines 30-32: What is the physical sense of the statement that dynamic range of k is smaller than of F10.7? What are practical conclusions from this statement? This should be explained, otherwise I don't see why is this statement needed.

This is similar to the previous point. The reduction in dynamic range indicates that k is only weakly dependent on the electron density changes associated with the change in f10.7 through the solar cycle. We have added an explanatory sentence referring back to the previous section.
"This, again, is indicative of the TEC dependent component of the residual error being largely modelled by the squared L1/L2 bending angle difference term in the correction."

p.6, line 11: "*In order the build the models*..." It
should be "in order to build the models".

Agreed. Text amended.

p.6, lines 20-25: It may be better to introduce "chi" here (rather than after equation 8) because "chi" is used in discussion of figures 11-13.

We do not think the symbol $\chi$ needs to be introduced until equation 8. However, to avoid any confusion, we have included the symbol in the captions of Figures 11-13.

p.6, lines 11-13; p.7, line 7: What is the reason for using two different sets, generated with the same "random drivers", for building and testing the k-model? I assume that the sets are statistically representative and the results are statistically significant. Will the results be substantially different with the use of one set for building and testing?

The aim is simply to have a statistically similar, but independent set of data for testing the models. Given the size of the test sets and the low complexity of the proposed model, we do not expect the model parameters to vary significantly if determined from one set or the other.

p.7, line 12: "... *residual errors are reduced to 2.2x$10^{-10}$ rad and 2.0x$10^{-9}$ rad respectively*."
The number to which something is reduced requires the number from which that something is reduced. Also, see comment to p.1, lines 16-17.

Agreed. Text has been amended to compare results of the uncorrected case to the model k case

Figure 9: Axes labels on all figures, except Figure 9, have units in parentheses. Thus "k (impact parameter 60 km)" is confusing. It should be "k (1/rad)". The title above the figure can be changed to "k value at 60 km above London".

Agreed. Figure has been amended.

Figure 16: I think, "left" and "right" are mixed up in the caption. Left is full histogram, while right is zoomed to highlight tails.

Agreed. Text amended.

---

## Referee Report (RR1)

Reviewer's comments on
"Improved model for correcting the ionospheric impact on bending angle in radio occultation measurements"
by  M. J. Angling , S. Elvidge, and S. B. Healy

I suggest only two technical improvements:
please define TECu as used in Fig. 6
please define solar flux units (sfu) as used in Figs. 9, 11-13

I recommend publishing the revised version of the paper.

---

## Referee Report (RR2)

The paper is revised and improved. The authors responded to all comments. The only remaining comment/question is about the height interval of interest 40-80 km.

From current author's response:

*In real data the corrected bending angles increase rapidly towards the surface. This means that the impact of any residual error becomes less insignificant below approximately 40 km. Furthermore, the VK94 correction assumes that the ray impact parameter/tangent height is below the ionosphere (i.e. the electron density is zero). Consequently, the main area of interest for estimation is between 40 and 80 km.*

While both statements are correct, none of them explains whether GPS RO ionospere-corrected (to 1st and 2nd orders) bending angles are useful for detection of climate signals at 80 km.

In [Danzer et al., 2013] (cited in the paper), there is a reference to [Ringer and Healy, 2008] (not cited in the paper). A decadal climate trend, projected into bending angle space, was estimated as 1.2 and 4 mcrad at heights 30 and 26 km. At those heights, the mean bending angle should be about 300-600mcrad. At 80 km, the mean bending angle should be about 0.5mcrad. What is an expected magnitude of climate trendin the bending angle space at 80 km? Can it be detectable with GPS RO even with the 2nd order correction of large-scale ionospheric effects?

I am not requesting response to this question at this time, by leaving it at the discretion of the authors.

I recommend the paper for publication.

---

## Author Response (AR2)

**Response to Reviewer's comments on**
**"Improved model for correcting the ionospheric impact on bending angle in radio occultation measurements"**

Responses in line below in red

**Reviewer 1:**
I suggest only two technical improvements:
please define TECu as used in Fig. 6
Definition included in Figure 6 caption

please define solar flux units (sfu) as used in Figs. 9, 11-13
Definition included in text (section 4.3) and in figure captions

**Reviewer 2:**
In real data the corrected bending angles increase rapidly towards the surface. This means that the impact of any residual error becomes less insignificant below approximately 40 km. Furthermore, the VK94 correction assumes that the ray impact parameter/tangent height is below the ionosphere (i.e. the electron density is zero). Consequently, the main area of interest for estimation is between 40 and 80 km.

While both statements are correct, none of them explains whether GPS RO ionosphere corrected (to 1st and 2nd orders) bending angles are useful for detection of climate signals at 80 km. In [Danzer et al., 2013] (cited in the paper), there is a reference to [Ringer and Healy, 2008] (not cited in the paper). A decadal climate trend, projected into bending angle space, was estimated as 1.2 and 4 mcrad at heights 30 and 26 km. At those heights, the mean bending angle should be about 300-600mcrad. At 80 km, the mean bending angle should be about 0.5mcrad. What is an expected magnitude of climate trend in the bending angle space at 80 km? Can it be detectable with GPS RO even with the 2nd order correction of large-scale ionospheric effects?

We thank the reviewer for his comment. Whilst the correction technique may work between 40 and 80km due to the reasons explained in the paper, we did not intend to suggest that climate studies would be well served by this approach at 80km.

[revised manuscript text omitted]